# Suppression of Chitin-Triggered Immunity by Plant Fungal Pathogens: A Case Study of the Cucurbit Powdery Mildew Fungus *Podosphaera xanthii*

**DOI:** 10.3390/jof9070771

**Published:** 2023-07-21

**Authors:** Nisrine Bakhat, Alejandra Vielba-Fernández, Isabel Padilla-Roji, Jesús Martínez-Cruz, Álvaro Polonio, Dolores Fernández-Ortuño, Alejandro Pérez-García

**Affiliations:** 1Departamento de Microbiología, Facultad de Ciencias, Universidad de Málaga, 29071 Malaga, Spain; nisrinebakhat@uma.es (N.B.); alejandravielbafdz@gmail.com (A.V.-F.); ipadilla@uma.es (I.P.-R.); jesusmcruz@uma.es (J.M.-C.); alvaropolonioescalona@gmail.com (Á.P.); aperez@uma.es (A.P.-G.); 2Instituto de Hortofruticultura Subtropical y Mediterránea “La Mayora”, Universidad de Málaga, Consejo Superior de Investigaciones Científicas (IHSM-UMA-CSIC), 29071 Malaga, Spain

**Keywords:** chitin-triggered immunity, effectors, *Podosphaera xanthii*, powdery mildews

## Abstract

Fungal pathogens are significant plant-destroying microorganisms that present an increasing threat to the world’s crop production. Chitin is a crucial component of fungal cell walls and a conserved MAMP (microbe-associated molecular pattern) that can be recognized by specific plant receptors, activating chitin-triggered immunity. The molecular mechanisms underlying the perception of chitin by specific receptors are well known in plants such as rice and *Arabidopsis thaliana* and are believed to function similarly in many other plants. To become a plant pathogen, fungi have to suppress the activation of chitin-triggered immunity. Therefore, fungal pathogens have evolved various strategies, such as prevention of chitin digestion or interference with plant chitin receptors or chitin signaling, which involve the secretion of fungal proteins in most cases. Since chitin immunity is a very effective defensive response, these fungal mechanisms are believed to work in close coordination. In this review, we first provide an overview of the current understanding of chitin-triggered immune signaling and the fungal proteins developed for its suppression. Second, as an example, we discuss the mechanisms operating in fungal biotrophs such as powdery mildew fungi, particularly in the model species *Podosphaera xanthii*, the main causal agent of powdery mildew in cucurbits. The key role of fungal effector proteins involved in the modification, degradation, or sequestration of immunogenic chitin oligomers is discussed in the context of fungal pathogenesis and the promotion of powdery mildew disease. Finally, the use of this fundamental knowledge for the development of intervention strategies against powdery mildew fungi is also discussed.

## 1. Introduction

Fungal pathogens are major threats to terrestrial plants. An estimated 30% of losses in annual global crop production are due to fungal infections before and after harvest [1]. It is widely accepted that the three most important plant fungal diseases are rice blast, gray mold, and wheat rusts, caused by *Magnaporthe oryzae*, *Botrytis cinerea*, and *Puccinia* spp., respectively [2]. The fungus *M. oryzae* has a destructive impact on rice, which is considered one of the primary sources of calories for half of the world’s population. *Botrytis cinerea* has a broad host range, causing serious damage both pre- and post-harvest. *Puccinia* spp. were already a severe disease concern, but their impact has grown with the *P. graminis* f. sp. *tritici* Ug99 race, which is a serious challenge to wheat production. Focusing on the most common plant fungal diseases, powdery mildews take the first position. Powdery mildew fungi form a telltale white, dusty coating on leaves, stems, and flowers. They affect several angiosperm plants, including apples, cereals, cucurbits, grapes, peas, and tomatoes, and ornamentals such as lilacs, phlox, daisies, and roses [3]. Among them, *Blumeria graminis* is the most important, affecting important agricultural crops such as wheat and barley [2].

Plants have defense mechanisms to protect themselves against pathogen invasions. These mechanisms can be divided into two categories: The first line of defense is the so-called “passive defense”, which include the cuticle, a waxy covering deposited on the plant surface, the plant cell wall, and anti-microbial substances such as enzymes, peptides, and secondary metabolites [4]. On the other hand, the “inducible defense” forms the second line of plant defense mechanisms, which include structural fortifications at the site of pathogen invasion, as well as the development and secretion of anti-microbial molecules at the site of infection but also in distant tissues [4]. Furthermore, a localized apoptosis-like hypersensitive cell death response may develop at the infection site, which can eventually lead to systemic acquired resistance (SAR), a long-lasting condition of induced immunity against a wide range of pathogens [5]. The first inducible defenses are launched by pattern recognition receptors (PRRs), which are cell surface receptors that recognize microbial-associated molecular patterns (MAMPs, also known as pathogen-associated molecular patterns or PAMPs) as non-self components, resulting in MAMP-triggered immunity (MTI, also known as PAMP-triggered immunity or PTI). This defense response consists of local cell wall fortifications, the generation of reactive oxygen species (ROS), and the synthesis and release of anti-microbial chemicals, all of which together will block most microbial invaders. Successful pathogens bypass MTI by using secreted effectors that actively disrupt host defenses, generating effector-triggered susceptibility (ETS) [6,7,8]. Plants can recognize various microbes via perception of conserved MAMPs such as lipopolysaccharides from Gram-negative bacteria, peptidoglycans from Gram-positive bacteria, and eubacterial flagellin [9]. Glucans, chitins, and proteins are derived from fungal cell walls; hence, chitin does not exist in plant cells, making it an excellent MAMP [10]. These chitin oligomers are released from fungal cell walls because of the activity of plant chitinases following a fungal attack [11]. They can be recognized by specific plant receptors, activating the plant response known as chitin-triggered immunity and pushing the plant to produce further defense mechanisms [12,13]. On the other hand, fungi are capable of creating defense mechanisms to prevent cell wall chitin digestion by plant enzymes or to interfere with chitin receptors or signaling in plants, which ends up promoting fungal colonization and disease development [14]. In this review, current models of chitin-triggered immune signaling and the fungal strategies to suppress chitin-triggered plant immunity, focusing on the cucurbit powdery mildew *Podosphaera xanthii*, will be presented.

## 2. Chitin-Triggered Plant Immunity

Fungal cell walls are dynamic structures essential for cell survival, morphogenesis, and pathogenesis [15]. Chitin, a homopolymer of β-(1,4)-linked N-acetyl-D-glucosamine (GlcNAc) and the second most abundant polysaccharide in nature after cellulose [16], is a crucial structural component of fungal cell walls. Chains of chitin associate in microfibrils in these cell walls, which are covalently attached to the primary component of fungal cell walls, (1,3)-glucan, forming a network with glycoproteins to provide the structural basis that confers stiffness to the cell wall. Other proteins and carbohydrates found in fungal cell walls include mannans, (1,6)-glucan, and (1,3)-glucan, which are part of the soluble matrix [17].

Chitin serves as an MAMP in plants, being recognized by lysin motif (LysM) PRRs localized on the plant cell surface (Figure 1) [6,18,19]. Its recognition prompts plants to secrete hydrolytic enzymes, such as chitinases and β-glucanases, to target fungal cell wall components, affecting fungal cell wall integrity. Two types of chitinases have been demonstrated to grant plants a defense against fungi in different ways: the first is exochitinases, which release chitin oligosaccharides that are recognized by host receptors to effectively trigger MTI, all while remaining in the apoplast and being unharmful to fungal growth [14,20,21]. The second, endochitinases, which are produced as a result of these reactions, accumulate in the vacuole and are released upon cellular collapse to operate as potent antifungals with an ability to hydrolyze fungal cell walls [22,23,24]. This activates further immune responses, including the accumulation of toxic compounds such as the production of reactive oxygen species (ROS), strengthening of plant cell walls by the aggregation of deposits of callose and lignin, and, in addition to ion fluxes, defense-related gene expression and even hypersensitivity reactions (Figure 1) [6,25,26,27].

Research into the molecular mechanisms of chitin detection and chitin-triggered immunity in plants has proliferated since the identification of a chitin receptor in rice (*Oryza sativa*) after a cross-linking of plasma membranes with radio-labeled chitin [28]. A 75 kDa protein was subsequently discovered, cloned, and named Chitin Oligosaccharide Elicitor-Binding Protein (CEBiP) [11]. CEBiP is a conventional receptor-like protein (RLP), holding an extracellular domain with two predicted Lysin motifs (LysMs) at the N terminus, which are known to bind bacterial peptidoglycan and fungal chitin, and a short membrane-spanning domain at the C terminus [29,30]. It was also shown to lack a cytoplasmic kinase domain, which suggests the need for other molecular elements to activate cellular signaling and cause chitin-triggered immunity [11]. Therefore, Chitin Elicitor Receptor Kinase-1 (OsCERK1), a LysM-containing cell surface receptor with a cytoplasmic kinase domain, was identified and demonstrated to be a partner molecule of OsCEBiP in rice [12]. Chitin-induced responses were reduced in cell lines with loss-of-function mutations of OsCEBiP or OsCERK1 generated by homologous recombinations, supporting the need for these molecules for chitin signaling and implying the creation of an OsCEBiP–OsCERK1 receptor complex [30,31]. Moreover, by affinity labeling with biotinylated chitin octamer (GN8-Bio) and precipitation assays with colloidal chitin, it was discovered that OsCERK1 lacked chitin binding abilities [32]. However, affinity labeling tests revealed that the GN8-Bio binding protein was entirely absent from the microsomal membranes of CEBiP knockout cell lines, demonstrating that CEBiP is the primary chitin receptor in rice [31].

On the other hand, five LysM receptor-like kinase 1 (LysM-RLKs) chitin elicitors are encoded in the genetic model plant *Arabidopsis thaliana*’s genome, with just one implicated in chitin detection, called AtCERK1 for chitin elicitor receptor kinase 1 or LysM RLK1 [13,33]. AtCERK1, an RLK, is formed of a transmembrane domain, three tandem LysMs in its ectodomain, and an intracellular serine/threonine kinase domain [13,33]. It was also demonstrated that chitin-induced AtCERK1 homodimerization and two additional LysM-RLKs (AtLYK4 and AtLYK5) are all required for its activation, with AtLYK5 mutations resulting in a significant decrease in the plant chitin response [34,35,36].

In the case of pathogenic fungal defense responses, CEBiP and CERK1 require longer chito-oligomers, such as octamers, to activate their response via heterodimerization (Figure 1) [37,38]. Shorter oligosaccharides, such as chitotetraose, may instead rely on monomeric receptors because half of this oligomer molecule (C4) fits inside each receptor’s LysM domain [10,34,39].

## 3. Fungal Strategies to Overcome Chitin-Triggered Immunity

Fungi have developed several strategies to subvert chitin-triggered host immunity in plants by either blocking chitin perception or signal transduction. The different strategies used by fungi to overcome chitin-triggered immunity are shown in Table 1 and Figure 2.

### 3.1. LysM and Non-LysM Containing Effectors

Many fungal pathogens encode LysM (also known as the carbohydrate-binding module CBM) domain-containing apoplastic effectors that outnumber plant chitin receptors containing the same domain (Figure 2A). Ecp6 from *Cladosporium fulvum* [42], Slp1 from *M. oryzae* [51], and RsLysM from *Rhizoctonia solani* [64] specifically bind to chitin oligomers and suppress chitin-induced immune responses in tomatoes, tobacco, rice cells, and leaves. By directly competing with a plant chitin receptor, CEBiP, for chitin binding, they attenuate ROS production triggered by chitin and inhibit the induction of defense-associated genes, therefore playing a role in evading the plant immune system by interfering with chitin perception and suppressing the host immune response. Similarly, *M. oryzae* secretes active chitinase MoChia1 in the apoplast that can bind to chitin and prevent its recognition by OsCERK1, suppressing the chitin-mediated immune response; this triggers the plant’s immune responses such as ROS burst, callose deposition, and MAPK signaling, but also the production of a protein called OsTPR1, which competes with chitin for binding to MoChia1, thereby re-establishing the activation of the immune response [50,73]. On the contrary, *Moniliophthora perniciosa* secretes the inactive chitinase MpChi to sequester immunogenic chitin fragments and maintain substrate binding specificity to interfere with the perception of plant chitin receptors and suppress chitin-triggered immunity [52]. Some fungal LysM effectors, including *Colletotrichum higginsianum*’s ChELP1 and ChELP2 [43] and *Verticillium dahlia*’s Vd2LysM [70], are thought to scavenge chitin fragments and compete with plant chitin receptors. By binding to both chitin polymers and oligomers, they protect the fungal cell wall and immediately veil chitin oligomers from plant perception. Notably, N-glycosylation of Slp1 is required for its protein stability and chitin affinity [74], whereas ChELP1 and ChELP2 are also N-glycosylating [43], indicating that N-glycosylation is generally required for fungal LysM effectors to achieve an extremely high affinity for chitin. Similarly, the arbuscular mycorrhizal fungus *Rhizophagus irregularis* and the endophytic fungus *Trichoderma atroviride* encode dual function LysM effectors RiSLM and Tal6 [65,66]. RiSLM protects fungal hyphae from hydrolysis by plant chitinases and suppresses chitin-triggered immune responses in Medicago roots. As for Tal6, it interferes with the plant’s immune response by binding to GlcNAc, a component of chitin, which could act as MAMPs. By doing so, Tal6 attenuates the plant’s defense response, allowing *Trichoderma* to establish itself in the roots without being fully detected or attacked by the plant. Moreover, the wheat Septoria tritici blotch (STB), *Zymoceptoria tritici* (also known as *Mycosphaerella graminicola*), secretes LysM effector proteins (Mg1LysM and Mg3LysM) to protect the hyphae from host hydrolytic enzymes, allowing the pathogen to establish and spread within the plant, noting that Mg3LysM, but not Mg1LysM, inhibit chitin-induced ROS production [72].

Non-LysM effectors are also used by fungal pathogens to defend their cell walls or obstruct plant chitin sensing (Figure 2B). Avr4- or Avr4-like effectors with a CBM14 domain can be secreted by *C. fulvum* and *Mycosphaerella fijiensis*, function as a cell wall protein, and specifically bind to chitin. This way, they protect the pathogen against plant chitinases and β-1,3-glucanases, allowing a successful colonization of the host [41,53]. In *Parastagonospora nodorum*, the recognition of the effector protein SnTox1 by the Snn1 receptor triggers defense responses in wheat, including programmed cell death (PCD), increased production of ROS, DNA laddering, and the up-regulation of antimicrobial proteins such as chitinases. However, it was also found that SnTox1, besides inducing a defense response, provides protection against chitinases that are up-regulated, suggesting that SnTox1 may counteract the plant’s defense mechanisms and facilitate the successful colonization of the pathogen [54]. To interact with chitin polymers and oligomers, *Verticillium nonalfalfae* releases apoplastic effectors, such as VnaChtBP, which contain a CBM18 domain capable of binding to chitin oligomers with a high affinity, enabling the fungus to evade both plant chitinases and chitin receptors [71]. In addition, the GPI-anchored proteins that are connected to the fungal cell wall in *M. oryzae* can serve in development, virulence, and immune evasion strategies by delaying the exposure of PAMPs such as chitin and β-1,3-glucans which are recognized by PRRs, helping the pathogen to suppress the plant’s immune response [49].

### 3.2. Chitin or Polysaccharide Deacetylases

To turn chitin into chitosan, many fungi can encode polysaccharide deacetylases to avoid chitin-induced plant immunity (Figure 2C). *V. dahlia*, *F. oxysporum*, and the endophytic *Pestalotiopsis* fungus use chitin deacetylases (VdPDA1, FovPDA1, and PesCDA, respectively) to convert chitin oligomers into immunogenically inactive chitosan oligomers that cannot be recognized by the plant’s chitin receptors, noting that partially acylated chitosan oligomers with at least five acetyl groups do trigger the production of ROS, which activates the plant immune system [46,55]. Moreover, the pathogen *Puccinia graminis* also secrets PgtCDA, which converts chitin into chitosan, as the invasive hyphae have been shown be adorned with chitosan rather than chitin [62]. On the other hand, a polysaccharide deacetylase (Pst_13661) from the fungal pathogen *Puccinia striiformis* f. sp. *tritici* (Pst) can bind to chitin, cellulose, and germ tubes of Pst and exhibits a deacetylase enzyme activity, which helps protect the fungal hyphae from degradation and reduces the generation of chitin fragments interfering with the plant’s recognition of chitin as a PAMP. As a result, the effector facilitates the infection and colonization of Pst in wheat plants [61].

### 3.3. Cytoplasmic Effectors

Fungal pathogens have also sneaked cytoplasmic effectors or other molecules into plant cells to disrupt chitin-triggered immunity (Figure 2D). To prevent chitin-induced ROS bursts, CoNIS1 and MoNIS1, two homologous effectors from *Colletotrichum orbiculare* and *M. oryzae*, respectively, can suppress BIK1 kinase activity, leading to the inhibition of BIK1–RbohD interactions and suppression of PAMP-triggered ROS generation, which is important for plant immune responses [44]. *M. oryzae* also uses the effector protein AvrPiz-t to target rice OsRac1, a protein involved in chitin-triggered ROS production, suggesting that AvrPiz-t may manipulate ROS production in host cells. AvrPiz-t has also been shown to suppress programmed cell death (PCD) and enhance necrosis, promoting the growth of *M. oryzae* [48]. Although *R. solani* may deliver the lipase-like effector AGLIP1, an ectopic expression of this effector in transgenic *Arabidopsis* plants led to the suppression of immune responses triggered by chitin. This suggests that AGLIP1 may have a broader role in manipulating the plant’s immune system and interfering with the recognition of chitin as a PAMP, enhancing disease development [63]. Moreover, the rice fungal pathogen *Ustilaginoidea virens* can release the tiny cysteine-rich effector SCRE2 to suppress chitin-triggered rice immunity and hypersensitive responses in the non-host plant *Nicotiana benthamiana*, meaning this protein is required for full virulence in *U. virens* [67]. It is also interesting to note that the fungus *Fusarium graminearum* produces a small RNA molecule (*Fg*-sRNA1) that can cross-species transport into wheat cells, silencing the TaCEBiP chitin receptor gene, weakening the plant’s resistance to this pathogen, and suppressing chitin-triggered immunity [75].

### 3.4. Chitinases and Other Mechanisms

A variety of potential secreted effectors with the chitin-binding domain, carbohydrate-binding module family 18 (CBM18), chitinase, and chitin deacetylase domains are encoded in the genome of the clubroot pathogen *Plasmodiophora brassicae*. Two of these proteins, PbChiB2 and PbChiB4, were shown to protect the spores of *P. brassicae* from being attacked by plant chitinases and capture chitin molecules released during the pathogen penetration into the host cells, preventing the plant’s immune system from recognizing and responding to chitin presence. They also inhibited chitin-triggered activation of the map kinase proteins MPK3 and MPK6 in rapeseed (Figure 2E) [56]. Moreover, numerous maize fungal pathogens, including *Bipolaris zeicola* [40], *Stenocarpella maydis* [40], and *F. verticilloides* [47], produce fungal chitinase-modifying proteins (CMPs) (Figure 2F). In the case of *B. zeicola* and maize, ChitA is a constitutively produced defense protein in healthy maize seeds. *B. zeicola* produces Bz-Cmp, an enzyme that catalytically modifies ChitA, converting it to a modified form called ChitA-m [40]. Similarly, *S. maydis* secretes Stm-Cmp, which efficiently modifies one of the ChitA forms, namely ChitA-F [40]. As for *F. verticilloides*, a study found that Fv-Cmp can truncate the seed chitinases ChitA and ChitB in maize [47]. The modification of ChitA by these fungal effectors can have different effects on the plant’s defense responses and on fungal infections. ChitA may function to inhibit the spread of fungal hyphae by degrading fungal cell wall chitin or by releasing chitin elicitors from fungal cell walls. This triggers plant defense responses, including the production of pathogen response proteins and the hypersensitive response, which is a form of programmed cell death involved in plant immunity. The modification of ChitA by these fungal effectors could serve to suppress direct growth inhibition or the induction of maize defense responses, promoting the successful colonization of the pathogen to establish infection [40,47]. On the other hand, *M. oryzae* can synthesize α-1,3-glucan, a non-degradable polysaccharide, to mask its cell wall surface during infection (Figure 2E). This plays a role in protecting the fungal cell wall from antifungal agents secreted by plants during infection and delays the release of PAMPs such as chitin and host defense responses [76]. Furthermore, *Ustilago maydis* secretes the fungalysin metalloprotease UmFly1 responsible for cleaving and deactivating maize chitinases, including ZmChiA, the specific chitinase in maize. UmFly1 targets chitinase at the chitin-binding domain (CBD), thereby reducing its efficiency in degrading fungal chitin. By cleaving and inactivating maize chitinases, UmFly1 helps *U. maydis* evade plant defense responses induced by chitin fragments from the fungal cell wall. This allows the fungus to establish and maintain a biotrophic interaction with the plant, promoting its virulence [68]. In contrast, *F. oxysporum* can secrete the metalloprotease FoMep1 and serine protease FoSep1, which act synergistically to cleave the tomato chitinases SlChi1 and SlChi13 at different locations in the N-terminus to increase virulence [45]. Similarly, when cotton plants are infected with *V. dahliae*, they secrete the apoplastic class IV chitinase Chi28, which cleaves fungal chitin, producing PAMPs that trigger immune responses. However, the fungal pathogen secretes a serine protease VdSSEP1, which can cleave and degrade Chi28, compromising the plant’s defense mechanism. However, it was also found that cotton CRR1 interacts with Chi28 in the apoplast, protecting it from degradation by VdSSEP1, ensuring Chi28′s stability and functionality, and playing a major role in the plant’s defense response to the pathogen [69].

In summary, fungi have developed unique or combined methods to compromise plant immunity induced by chitin. Some counterstrategies, such as coating chitin with fungal proteins to prevent plant digestion or perception, seem to be very common in fungi. Future research on fungal cytoplasmic effectors is projected to flourish, which will lead to the discovery of further action mechanisms of fungi.

## 4. The Powdery Mildew Fungi and the Cucurbit Pathogen *Podosphaera xanthii* as a Model Species

Powdery mildew fungi are pathogens belonging to the class *Ascomycota*, order *Erysiphales*, and family *Erysiphaceae*. They can infect and cause disease in a wide range of monocotyledonous and dicotyledonous species worldwide, including economically significant crops like barley, wheat, cereals, grapevines, tomato, several vegetables, and ornamental species [3,77]. They are easily recognized when infecting their hosts because their typical signs on plants are a whitish, talcum-like, powdery fungal growth on the petioles, stems, and occasionally fruits [78] (Figure 3A). Although cultural and biological methods could minimize the risk of powdery mildew infection, they do not provide adequate plant protection. As a result, chemical control, including the use of fungicides from several chemical groups, is the most effective method in practice. Unfortunately, some of the most economically important powdery mildew species are considered high-risk pathogens, capable of developing resistance to numerous chemical classes a few years after use [79].

To develop and complete their life cycle, powdery mildews must feed on living host cells to survive, being considered as obligate biotrophic. For this, they have evolved specialized structures of parasitism termed haustoria (Figure 3B), whose primary roles are the intake of nutrients from the host [80,81,82,83,84] and the release of effectors into plant cells [85,86,87,88,89,90,91,92,93,94,95]. These effectors are defined as substances that alter the physiological structure and metabolism of the host cell, modulate the plant’s defense response, and protect the integrity of the fungus to allow its colonization and proliferation [96]. Although both asexual and sexual stages are present in the pathogen’s life cycle, the asexual cycle seems to be primarily responsible for pathological damage (Figure 3C). Following a conidium’s contact with a susceptible host, the asexual life cycle can be divided into three different stages: (i) primary appressorium formation and penetration of the cuticle and cell wall; (ii) primary haustorium formation; and (iii) the growth of a mat of hyphae (with the formation of secondary appressoria and haustoria) and the differentiation of conidiophores and conidia [93,97,98,99,100]. In the event of a sexual cycle, two compatible mating type hyphaes need to be united to create a fruiting body termed as a chasmothecium, which holds one or more asci containing sexual spores or ascospores [101]. Although the exact pathogenesis of the infection caused by the ascospores needs to be clarified, it is assumed to be very similar to those caused by conidia [79,102]. Due to its lifestyle, they cannot be cultured on artificial culture media, which complicates their manipulation under laboratory conditions, a fact that has greatly restricted research from a genetic and molecular point of view in comparison to other filamentous plant pathogens [100].

Among powdery mildew, *P. xanthii* stands out as the main causal agent of cucurbits powdery mildew disease and one of the most serious limiting factors for cucurbit production worldwide [103,104]. Like other powdery mildew species, *P. xanthii* haustorium resembles a bulb attached to an epiphytic fungal hypha and exhibits the usual tubular elongations known as lobules [105,106,107]. It contains microscopic vesicles and multivesicular bodies (MVBs) and, in addition, is gradually encased in deposits of plant cell wall polymers because of its contact with the host plant’s cells [92]. Due mostly to challenges in isolating powdery mildew haustoria, comprehensive molecular studies of these structures are currently rare. However, a lot of work has been put into determining the molecular bases of *P. xanthii*’s pathogenicity and biology in general. This has led to the development of essential resources such as the epiphytic and haustorial transcriptomes [108,109,110] and first draft genomes [111,112], as well as RNA-seq analyses of the initial stages of infection in melons and other hosts [113,114,115]. The first *P. xanthii* effectors were found due to the development of methods for the functional investigation of *P. xanthii* genes, including different transformation and RNA interference (RNAi) silencing protocols [116,117,118,119,120]. Despite this significant advancement, more investigations into *P. xanthii* are still required to find novel genes and pathways that enable us to comprehend the unique characteristics of this significant cucurbit pathogen and, ultimately, to create new management methods.

## 5. Countermeasures of *P. xanthii* for Subverting Chitin Signaling

Chitin is abundantly present in the cell walls of *P. xanthii*, either in epiphytic structures such as primary and secondary hyphae or in endophytic fungal cells, the haustoria (Figure 4). Due to *P. xanthii*’s obligate biotrophic lifestyle, the fungus needs to cope with PTI (PAMP-triggered immunity), a defense reaction brought on by fungal contact with the plant cells, the immunity elicited by chitin oligomers being particularly intense. For a successful infection, it is believed that an arsenal of *P. xanthii* effectors are required during penetration to downregulate chitin-mediated PTI signaling. Some of these effectors have been found in model biotrophs like *B. graminis* f. sp. *hordei*, *Cladosporium fulvum*, and *U. maydis* [121].

In previous studies, the epiphytic transcriptome of *P. xanthii* was sequenced and annotated, determining the presence of 138 proteins, including 53 *Podosphaera* effector candidates (PECs). These proteins were identified based on the presence of a predicted signal peptide and the lack of functional annotations [108]. Subsequently, host-induced gene silencing (HIGS) was used to uncover the genes involved during the initial plant–pathogen interactions [118]. Several PEC-encoding genes were chosen to be investigated, revealing that six of them—PEC007, PEC009, PEC019, PEC032, PEC034, and PEC054—are necessary for *P. xanthii* pathogenicity, as evidenced by a decreased fungal growth and an increased hydrogen peroxide production by host cells. Additionally, the biochemical activity of three of these effectors, PEC019, PEC032, and PEC054, was demonstrated, supporting the computational predictions and revealing new roles for powdery mildew effectors, including phospholipid-binding protein (PEC019/PxPLBP1), a-mannosidase (PEC032/PxMLE1), and cellulose-binding protein (PEC054/PxCLBE1). More intriguingly, these effectors are extensively dispersed in phytopathogenic fungi, according to BLAST searches, pointing to new targets for fungal effectors, including host–cell glycosylation, host–cell plasma membranes, and damage-associated molecular pattern-triggered immunity [119]. Woth respect to the way that *P. xanthii* avoids chitin recognition to deal with chitin resistance, this pathogen has developed several strategies such as binding, breaking, or modifying immunogenic oligomers, as described below and represented in Figure 5 and Table 1.

### 5.1. Degradation of Chitin Oligomers

Chitinases are well-known fungal proteins that play a role in cell wall remodeling during growth as well as competitive interactions with other fungi [122]. Chitinases and chitinase-like proteins have been found or computationally predicted in numerous powdery mildew genomes [83,87,123,124]. A recent study of *P. xanthii* revealed the involvement of a new family of secreted proteins with unclear functions [57,108]. According to RNAi silencing assays, protein modeling, protein–ligand predictions, enzymatic assays, and protein localization studies, these proteins have been characterized as possessing intrinsic chitinase activities, enabling them to break down immunogenic chitin oligomers released from fungal cell walls by plants chitinases at pathogen penetration sites. These effectors disrupt the integrity of fungal structures and modify the chitin-derived signals perceived by the plant’s immune system, preventing the activation of chitin-triggered immunity [57]. As a result, these proteins have been designated as effectors with chitinase activity (EWCAs) (Figure 5A) which differ mostly in their C-terminal regions that, interestingly, contain a low-complexity region (LCR) abundant in alanine residues linked to gene growth and diversification [125]. The reduced levels of chitin signals by EWCA activity leads to the attenuation of immune responses, as was determined after using various staining techniques to visualize callose deposits, which is indicative of cell wall strengthening, and hydrogen peroxide, which is a marker of ROS accumulation [41,51]. The silenced samples showed greater accumulation of callose deposits compared to the control group, which only exhibited weak yellow fluorescence in penetrated cells and small yellow spots at penetration points [57]. Similarly, there was a significant increase in the accumulation of brown precipitates, indicative of H_2_O_2_ production, in the plant cells of the silenced samples processed with the 3,3′-diaminobenzidine according to the DAB uptake method [57,118]. In terms of fungal development, the control group had large colonies with a high number of widely spaced haustoria per colony. In contrast, the silenced colonies were smaller and had a low number of haustoria, but the haustoria were more densely packed. In addition, fungal biomass quantification using qPCR showed a clear reduction in fungal growth after silencing the EWCA genes. Overall, the study demonstrated that different effector candidates within the EWCA gene family have varying effects on plant responses, suggesting functional diversity among these genes [57]. Additionally, similar genes were discovered in the genomes of numerous fungal pathogens, indicating that these effectors play an important role in fungal pathogenesis [57].

The *P. xanthii* haustorial transcriptome has also made it possible to discover the effector candidate PHEC27213, later known as PxLPMO1, which is the most highly expressed, haustorium-specific, putative secreted protein [110]. Different computational predictions showed that PxHEC27213′s protein folding was similar to a lytic polysaccharide monooxygenase (LPMO) (Figure 5B) that included a conserved histidine brace, but it had low sequence similarity with LPMO proteins and displayed a putative chitin-binding domain that differed from the canonical carbohydrate-binding module [58]. However, binding and enzymatic experiments revealed that this protein functions as an LPMO and could bind and catalyze colloidal chitin and chito-oligosaccharides generated by plant endochitinases during the growth of haustoria, evading chitin perception by the host plant and permitting the development of haustoria inside plant epidermal cells. Furthermore, to validate its role, an *Agrobacterium tumefaciens*-mediated host-induced gene silencing (ATM-HIGS) assay was conducted. The efficacy of ATM-HIGS was confirmed by the decreased transcript levels of PxLPMO1 during gene silencing experiments. After silencing PxLPMO1, the development of *P. xanthii* was significantly altered and delayed compared to the negative control. Additionally, there was a strong accumulation of H_2_O_2_, indicating the activation of chitin-triggered immunity in PxLPMO1-silenced tissues. Fungal growth quantification through haustorial counts and qPCR showed a drastic reduction in fungal development in PxLPMO1-silenced melon cotyledons [58]. To further confirm the role of PxLPMO1 in preventing chitin-triggered immunity, additional RNAi silencing experiments were conducted, including co-silencing of the melon chitin receptor kinase gene *CmCERK1*. Co-silencing *PxLPMO1* and *CmCERK1* restored the normal phenotype, with normal *P. xanthii* development and a low level of reactive epidermal cells. Fungal growth quantification and hydrogen peroxide production analyses supported these observations. Overall, the plant’s response to this fungal attack involved the activation of chitin-triggered immunity, suggesting that PxLPMO1 plays a crucial role in preventing the activation of the immune system [58].

### 5.2. Chitin Deacetylation

*P. xanthii* has developed another strategy to overcome chitin detection by the conversion of cell wall chitin into chitosan by chitin deacetylase (CDA; Figure 5B). Given the vast conservation of CDA in fungi, it is expected that deacetylation of chitin oligomers to avoid host recognition by chitin receptors is a widespread and conserved approach of plant pathogenic fungi to host survival [46]. This enzyme catalyzes the hydrolysis of the N-acetamido group in the N-acetylglucosamine units of chitin to produce chitosan, a poor substrate for chitinases and a molecule with a significantly lower elicitor activity than chitin [14,126]. In a previous study, the role of chitin deacetylase (CDA) in the pathosystem *P. xanthii* melon and its interaction with the plant’s immune response were investigated. Two transcripts of the *P. xanthii* CDA gene (*PxCDA1* and *PxCDA2*) were identified, and RNAi silencing experiments were conducted to reduce their expression [59]. The silencing of these transcripts resulted in a significant decrease in fungal growth and a concomitant increase in hydrogen peroxide production, suggesting the rapid activation of plant defense mechanisms against the pathogen. To further confirm the activation of chitin-triggered immunity, the same experiments were conducted with simultaneously silencing of the *PxCDA* gene and the plant chitin elicitor receptor kinase gene *CmCERK1*. In co-silenced tissues, fungal growth was fully restored and the production of hydrogen peroxide was considerably reduced. This confirms that the response activated in the plant after *PxCDA* silencing is indeed chitin-triggered immunity, and the interaction between CDA and the plant’s immune receptors is important for disease development [59]. Moreover, treatment with carboxylic acids, such as ethylenediaminetetraacetic acid (EDTA), a well-known CDA inhibitor, effectively suppressed powdery mildew disease, showing that CDA is a promising target for controlling phytopathogenic fungi and EDTA could be a starting molecule for fungicide design. In addition, EDTA treatment demonstrated efficacy in controlling other fungal diseases in strawberry and orange fruits caused by *B. cinerea* and *Penicillium digitatum*, respectively [59]. However, high concentrations of EDTA had a phytotoxic effect, hampering host responses and allowing fungal growth on the compromised tissue. In this regard, in a recent study, novel fungicidal compounds with the potential to inhibit CDA were identified using a computer-aided drug design approach called molecular topology [127]. These results indicate that CDA plays a major role in powdery mildew virulence and suggest that interfering with mechanisms of inhibition of chitin-triggered immunity could be a novel strategy for powdery mildew control [59].

### 5.3. Binding of Chitin Oligomers

In previous studies, effector genes containing LysM domains were found to be absent in the genomes of powdery mildew fungi [21], which was surprising, as suppression of chitin-triggered immunity should be essential for the survival in the host of these biotrophic fungi. In particular, this fact has been recently confirmed for *P. xanthii* as a result of analyses of three independent genomes [111,112,128], raising the question of whether a similar function to that of the LysM effectors may be performed by other proteins in these fungi. Perhaps the answer to this question lies in a recent study [60]. In this work, *PxCDA3*, a remarkably brief CDA transcript, was identified in *P. xanthii*. This transcript appeared to encode a shortened form of CDA because of an alternative splicing of the *PxCDA* gene, which retained the carbohydrate-binding module but lost most of the chitin deacetylase activity domain, suggesting a potential ability to bind chitin oligomers. Experiments with the recombinant protein demonstrated its capacity to bind to chitin oligomers and prevent the activation of chitin signaling, and localization studies using fluorescent fusion proteins showed that PxCDA1 is restricted to the fungal cell wall, while PxCDA3 is found in plant papillae. These are structures that form at pathogen penetration sites, containing high levels of chitin due to the activity of plant chitinases that break down chitin fragments released by the pathogen [57]. This suggesting PxCDA3′s role in scavenging chitin and preventing the activation of chitin-triggered immunity due to its chitin-binding capabilities similar to those of *C. fulvum* Ecp6 or *M. oryzae* Slp1 proteins [7,14,21,51,60]. Protein coding by PxCDA3 was proposed as a new fungal chitin-binding effector and designated CHBE (Figure 5A). This protein accumulates when plant chitinases are highly active during the early stages of infection, as *P. xanthii* uses different effectors at precise times and locations to disarm chitin signaling. In addition, the presence of cysteine residues in PxCHBE was significant. Cysteine residues can form disulfide bridges, which provide stability and protection against plant proteases [129]. This feature is characteristic of secreted proteins and allows PxCHBE to withstand the hostile environment in the plant and effectively perform its function. The plant’s immune response is a complex interplay of various defense mechanisms, including chitinases, antimicrobial peptides, ROS, and hormonal signaling pathways [27]. However, we can see that pathogens like *P. xanthii* evolving effector proteins such as PxCHBE to evade or defeat these immune responses allows them to establish successful infections, suppressing the activation of chitin-triggered immunity and promoting the growth and development of the fungus within the plant tissues. The authors also suggest that alternative splicing may be an evolutionary pathway for the emergence of new virulence factors in fungi, and in this case, powdery mildew fungi have evolved chitin-binding proteins that are involved in the manipulation of chitin-triggered immunity via chitin sequestration. These findings reinforce the idea that the evolution of molecular mechanisms to disarm the activation of chitin-triggered immunity is required for the successful colonization of plant habitats by fungi, especially haustorium-forming fungal pathogens.

Overall, the discovery of these effectors has provided insights into the complex interplay between fungal pathogens and plant immune systems. They play a critical role in suppressing chitin-triggered immunity by degrading, modifying, or binding to chitin molecules and intervening with chitin-derived immune signals. Understanding the mechanisms underlying the plant response to these fungal effectors is essential for developing strategies to enhance plant resistance against fungal pathogens and improve crop protection in agriculture.

## 6. Conclusions

Chitin-triggered immunity is a potent defensive response of plants against fungi, and therefore phytopathogenic fungi have evolved mechanisms to overcome this immunity. In the case of *P. xanthii*, these mechanisms come into play at two critical moments: during the penetration of the pathogen into the papilla and during the formation of the haustorium. As shown in our studies, any perturbation of these mechanisms leads to a rapid activation of chitin signaling [57,58,59,60], which opens the door to design novel intervention strategies against the pathogen. Currently, the management of powdery mildew diseases is based on two main control methods: the use of resistant cultivars and the application of fungicides. Although safer alternatives to chemicals, such as inorganic, organic, and biological control products, are also available, the application of fungicides is still the primary method for managing powdery mildews in many crops, which frequently results in the development of resistance to the most frequently used mildewcides [79]. A notable example of this issue is the cucurbit powdery mildew agent *P. xanthii*, which consistently develops resistance and swiftly renders several systemic fungicides useless. In this case, new technologies should be created to assist in achieving the intended aim of sustainability, by either breeding enhanced plant varieties or creating new, safer pesticides. In addition to natural products such as botanical and microbial materials, the biopesticide category now includes RNAi technology such as dsRNA. This technology adapts endogenous gene expression in plants to target pest and pathogen genes both inside plants (host-induced gene silencing, HIGS) and as topical applications (spray-induced gene silencing, SIGS) [120,130]. As mentioned above, chitin signaling manipulation in *P. xanthii* and other fungal pathogens is a complicated and carefully regulated mechanism that is mandatory for fungal pathogenesis [14,57]. Therefore, this example set of genes would be an ideal target for controlling powdery mildew in the future using the corresponding dsRNAs to promote the subsequent activation of chitin-triggered immunity. This control strategy is currently under examination in our laboratory.

## Figures and Tables

**Figure 1 jof-09-00771-f001:**
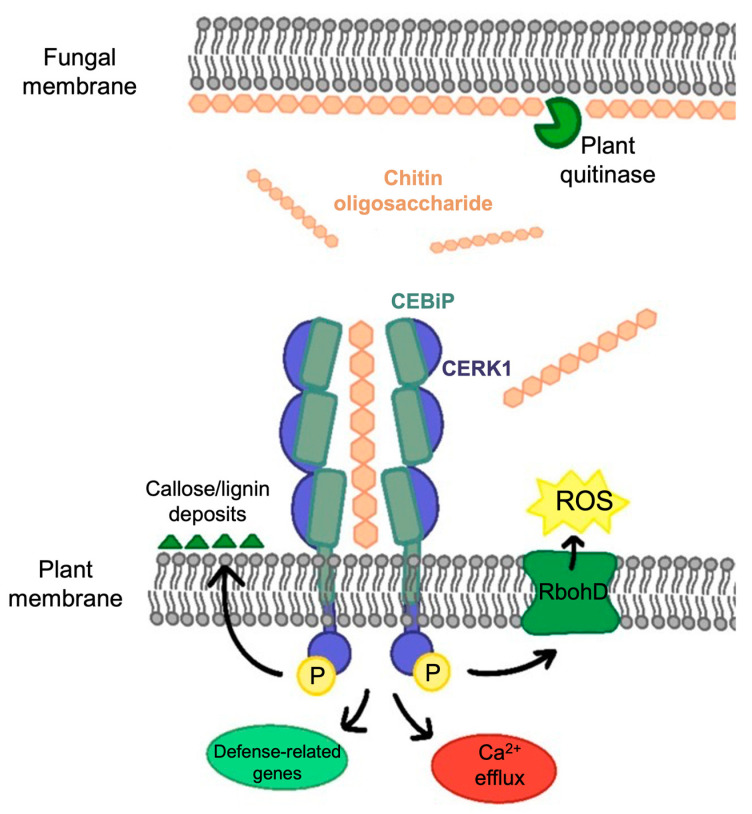
Host-secreted chitinases release chito-oligosaccharides from the fungal cell wall into the apoplastic space that are bound by the plant LysM containing receptor-like protein (RLP) CEBIP. Subsequently, the chitin-elicited receptor kinase 1 (CERK1) is recruited, which in turn leads to phosphorylation of its intracellular kinase domain. Activated CERK1 subsequently activates a cascade that triggers expression of defense genes, production of reactive oxygen species (ROS), Ca^2+^ efflux, and strengthening of plant cell walls by the aggregation of callose and lignin deposits.

**Figure 2 jof-09-00771-f002:**
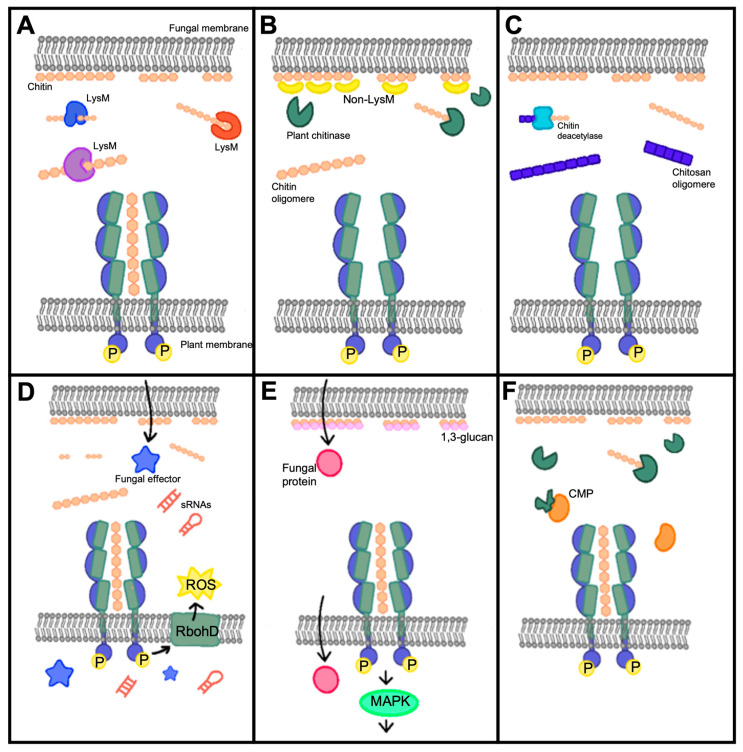
Fungal strategies to overcome chitin-triggered immunity. (**A**) Fungi secrete encode LysM domain-containing apoplastic effectors with high affinity to chitin oligomers to outnumber plant chitin receptors and block chitin perception. (**B**) Fungi also secrete non LysM effectors that can bind to cell wall chitin and stop its degradation by plant chitinases. (**C**) Fungi deacetylate chitin to chitosan to escape chitin-triggered immunity. (**D**) Fungi can sneak cytoplasmic effectors or other molecules such as sRNAs into plant cells to suppress chitin-induced reactive oxygen species (ROS). (**E**) Fungi accumulate nondegradable α-1,3-glucan on cell wall chitin to block its generation by plant chitinases, and also produce proteins that inhibit chitin-triggered activation of the map kinase proteins. (**F**) Fungi can produce chitinase-modifying proteins (CMPs) that can degrade extracellular chitinases.

**Figure 3 jof-09-00771-f003:**
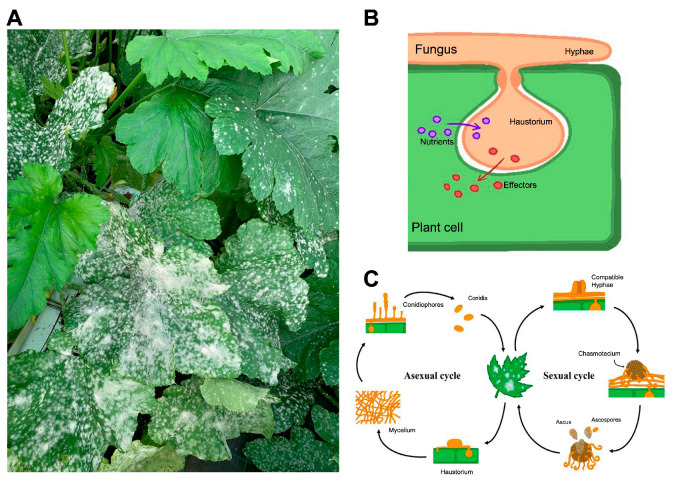
Powdery mildew fungi. (**A**) Typical symptoms of powdery mildew in cucurbits. (**B**) Schematic representation of the haustorium of powdery mildew fungi and its main functions. (**C**) A typical life cycle of a powdery mildew fungus (Taken from [79]).

**Figure 4 jof-09-00771-f004:**
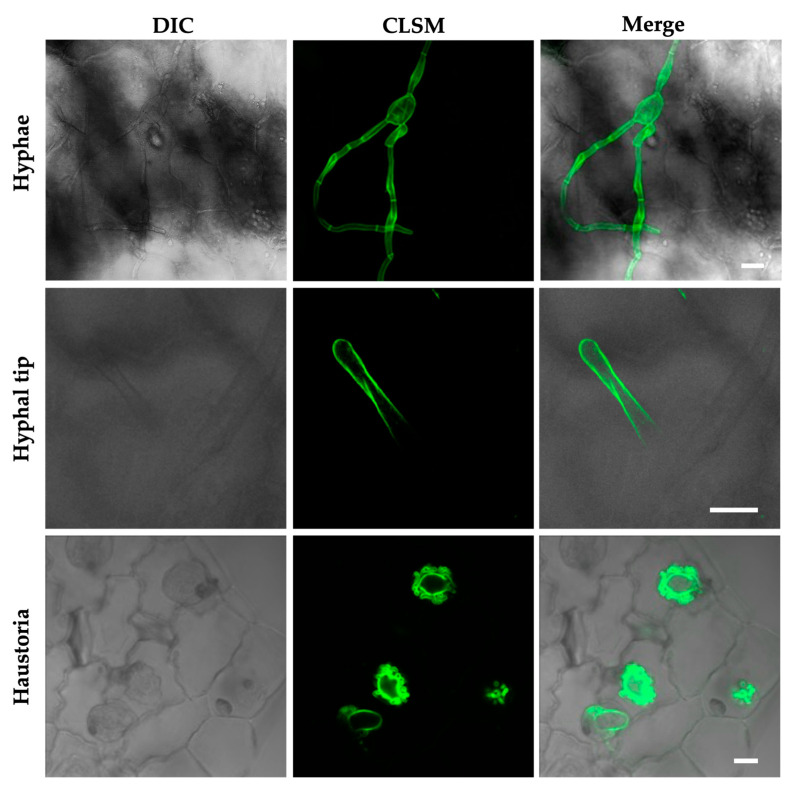
Visualization of chitin present in the cell walls of the developmental structures of *P. xanthii.* Melon leaves were inoculated with *P. xanthii* conidia and the presence of chitin in fungal structures was visualized using the chitin-specific staining wheat germ agglutinin (WGA) conjugated Alexa Fluor 488^®^ and confocal laser scanning microscopy (CLSM) [95]. DIC (differential interference contrast). CLSM images of *P. xanthii* hyphae, the hyphal tip, and haustoria are shown. Fluorescence signals indicate the presence of chitin in fungal cell walls. Bars: 20 µm (hyphae); 5 µm (hyphal tip); 10 µm (haustoria).

**Figure 5 jof-09-00771-f005:**
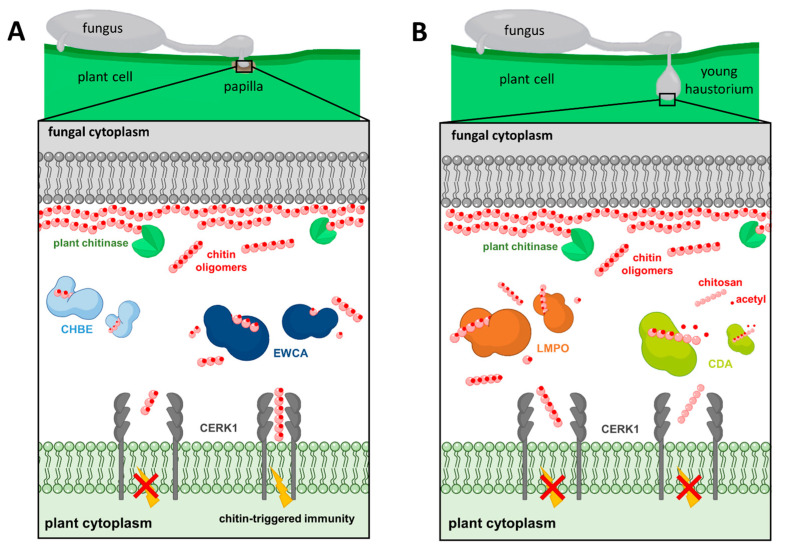
*P. xanthii*’s strategies to subvert chitin signaling. (**A**) EWCAs and CHBE are released by the fungus at the penetration site to break down or bind immunogenic chitin oligomers, respectively, blocking the dimerization of CERK1 and thus suppressing the activation of chitin-triggered immunity. (**B**) During the development of the haustorium, LMPO catalyzes chito-oligosaccharides into smaller molecules, whereas CDA converts cell wall chitin fragments into chitosan, thus evading chitin perception by the host plant.

**Table 1 jof-09-00771-t001:** Effectors and strategies to overcome chitin-triggered immunity employed by fungi.

Pathogen	Plant Host	Effector	Strategy	Function	References
*Bipolaris zeicola*	Maize	Bz-Cmp	Chitinase-modifying protein	Converts the *Zea mays* seed chitinase ChitA into a modified form: ChitA-m	[40]
*Cladosporium fulvum*	Tomato	Avr4	Non-LysM-containing effector	Associates with cell wall chitin to prevent hydrolysis by plant chitinases	[41]
		Ecp6	LysM-containing effector	Associates with chitin oligomers to prevent chitin perception by plants	[42]
*Colletotrichum higginsianum*	*Arabidopsis thaliana*	ChELP1 and ChELP2	LysM-containing effectors	Binds to both chitin polymers and oligomers to protect fungal cell walls from plant digestion and mask chitin oligomers to avoid plant perception	[43]
*Colletotrichum orbiculare*	*Nicotiana benthamiana*	CoNIS1	Cytoplasmic effector	Inhibits the kinase activity of BIK1 and BIK1–RbohD interaction to disrupt chitin-induced ROS burst	[44]
*Fusarium oxysporum*	Tomato	FoMep1 and FoSep1	Proteases	Cleaves tomato chitinases to promote virulence	[45]
	Cotton	FovPDA1	Polysaccharide deacetylase	Transforms chitin oligomers into immunogenically inactive chitosan oligomers to circumvent chitin-induced plant immunity	[46]
*Fusarium verticillioides*	Maize	Fv-Cmp	Chitinase-modifying protein	Truncates the plant chitinases ChitA and ChitB	[47]
*Magnaporthe oryzae*	Rice	AvrPiz-t	Cytoplasmic effector	Targets the host ubiquitin-proteasome system to manipulate plant defenses	[48]
		GPI-anchored proteins	Non LysM effectors	Works as a camouflage for chitin polymers to help evade plant digestion	[49]
		MoChia1	LysM-containing effector	Sequesters chitin oligomers and hinders the perception by plant chitin receptors	[50]
		MoNIS1	Cytoplasmic effector	Harnesses the effector protein AvrPiz-t to interfere with chitin-induced ROS bursts by targeting the rice OsRac1	[44]
		Slp1	LysM-containing effector	Associates with chitin oligomers to prevent chitin perception by plants	[51]
*Moniliophthora perniciosa*	Cacao	MpChi	LysM-containing effector	Prevents chitin-triggered immunity by sequestering immunogenic chitin fragments	[52]
*Mycosphaerella fijiensis*	Tomato	Avr4	Non-LysM-containing effector	Associates with cell wall chitin to prevent hydrolysis by plant chitinases	[53]
*Parastagonospora nodorum*	Wheat	SnTox1	Non-LysM-containing effector	Shields fungal cell walls from plant chitinases	[54]
*Pestalotiopsis sp*	Rice	PesCDA	Chitin deacetylase	Transforms chitin oligomers into immunogenically inactive chitosan oligomers to circumvent chitin-induced plant immunity	[55]
*Plasmodiophora brassicae*	Rapeseed	PbChiB2 and PbChiB4	Chitinases	Binds to spores and chitin oligomers to suppress chitin-triggered activation of the map kinase proteins MPK3 and MPK6 in the host	[56]
*Podosphaera xanthii*	Melon	PxEWCAs	Chitinases	Released during the pathogen penetration sites to break down immunogenic chitin oligomers, preventing the activation of chitin-triggered immunity	[57]
		PxLPMO	Monocopper enzyme	Binds and catalyzes colloidal chitin and chito-oligosaccharides generated by plant endochitinases during the growth of haustoria	[58]
		PxCDA	Chitin deacetylase	Converts chitin to chitosan	[59]
		PxCHBE	Chitin-binding effector	Binds to chitin oligomers and prevents the activation of chitin signaling	[60]
*Puccinia striiformis*	Wheat	Pst_13661	Polysaccharide deacetylase	Modifies chitin polymers rendering them recalcitrant to plant digestion	[61]
*Puccinia graminis*	Cereal	PgtCDA	Chitin deacetylase	Converts chitin to chitosan	[62]
*Rhizoctonia solani*	Rice	AGLIP1	Cytoplamic effector	Suppresses chitin-induced defense gene activation	[63]
	Sugarbeet	RsLysM	LysM-containing effector	Associates with chitin oligomers to prevent chitin perception by plants	[64]
*Rhizophagus irregularis*	*Medicago truncatula*	RiSLM	LysM-containing effector	Protects the hyphae from plant degradation and disguises chitin oligomers from plant detection	[65]
*Stenocarpella maydis*	Maize	Stm-Cmp	Chitinase-modifying protein	Modifies ChitA protein to ChitA-F	[40]
*Trichoderma atroviride*	*Arabidopsis thaliana*	Tal6	LysM-containing effector	Protects the hyphae from plant degradation and disguises chitin oligomers from plant detection	[66]
*Ustilaginoidea virens*	Rice	SCRE2	Cytoplasmic effector	Perturbs chitin-triggered immunity	[67]
*Ustilago maydis*	Maize	UmFly1	Protease	Truncates the maize chitinase ZmChiA	[68]
*Verticillium dahlia*	Cotton	SSEP1	Protease	Decapitates the cotton chitinase Chi28	[69]
		VdPDA1	Polysaccharide deacetylase	Transforms chitin oligomers into immunogenically inactive chitosan oligomers to circumvent chitin-induced plant immunity	[46]
	Tomato	Vd2LysM	LysM-containing effector	Binds to both chitin polymers and oligomers to protect fungal cell walls from plant digestion and masks chitin oligomers to avoid plant perception	[70]
*Verticillium nonalfalfae*	Common hop	VnaChtBP	Non-LysM-containing effector	Interacts with both chitin polymers and oligomers, allowing the fungus to escape from both plant chitinases and chitin receptors	[71]
*Zymoseptoria tritici*	Wheat	Mg1LysM and Mg3LysM	LysM-containing effectors	Protects fungal hyphae against host chitinases	[72]

## Data Availability

Not applicable.

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
