# Peer review of "Suppression of Chitin-Triggered Immunity by Plant Fungal Pathogens: A Case Study of the Cucurbit Powdery Mildew Fungus Podosphaera xanthii"

_jof, 2023, doi:10.3390/jof9070771_

Round 1
Reviewer 1 Report
This review focuses on how fungal infections, particularly Podosphaera xanthii, decrease chitin-triggered immunity in cucurbits, as the abstract indicates. Mentioning the fungal effectors involved in modifying, degrading, or sequestering chitin oligomers demonstrates a thorough comprehension of the subject. The abstract also discusses how this knowledge affects powdery mildew intervention options, which boosts the review's practicality.
Although the abstract provides a solid overview of the topic and presents the key components of the review, it needs minor improvement. It could be improved by including specific details on the current understanding of chitin-triggered immune signaling and fungal strategies, stating clear objectives or research questions, and limitations and gaps. These changes would improve the abstract and engage readers. There are also a few grammatical errors throughout the text, such as:
….two lines of defense: passive and inducible defenses…
Remove: for that (……plant immunity, using various strategies such as preventing…..)
…in the modification, degradation, or sequestration…
Introduction
….the rice blast, the gray mold, and the wheat rust, caused by Magnaporthe oryzae, Botrytis cinerea, and Puccinia spp., respectively [2].
…Ug99 race, which is a serious challenge to wheat production.
….fungal diseases, powdery mildews take the first position.
…a telltale white, dusty coating on leaves, stems, and flowers.
They affect several angiosperm plants, including apples, cereals, cucurbits, grapes, peas, tomatoes, and ornamentals such as lilacs, phlox, daisies, and roses [3].
…is the most important, affecting important…
Plants have defense mechanisms to..
..into two categories: The first line..
.."passive defenses", which include the cuticle, a waxy covering deposited on the plant surface, the plant cell wall, and anti-microbial substances such as enzymes, peptides, and secondary metabolites [4]. On the other hand, "inducible defense" forms the second line of plant defense mechanisms, which include…
..the infection site, which can eventually lead..
…as non-self components, …
….recognize various microbes via the perception of conserved MAMPs such as lipopolysaccharides from Gram-negative bacteria, peptidoglycans from Gram-positive bacteria, and eubacterial flagellin [9].
hence, chitin does not exist in….
….immunity and pushing the plant to produce further defense…
….capable of creating defense….
…by plant enzymes or to interfere with….
…mildew Podosphaera xanthii, will be presented.
2. Chitin-triggered plant immunity
..accumulate into the vacuole and are released upon…
..defense-related gene expression, and even hypersensitivity reactions (Figure 1)
Therefore, Chitin Elicitor Receptor Kinase-1 (OsCERK1), a …
…recombination, supporting the need for these..
3.1. LysM and non-LysM containing effectors
….in tomato, tobacco, rice cells, and leaves.
….callose deposition, and MAPK signaling,…
…Moniliophthora perniciosa secretes the inactive chitinase….
..the perception of plant chitin receptors and suppress….
….and Verticillium dahlia’s Vd2LysM [47], are thought to scavenge chitin fragments and compete with plant chitin receptors. By binding to both chitin polymers and oligomers, they protect the fungal cell wall….
….ChELP1 and ChELP2 are also N-glycosylated [46], indicating that N-glycosylation is…..
…Mycosphaerella graminicola), secretes LysM….
6. Conclusions
…at two critical moments: during the…
A notable example of this issue is the cucurbit powdery mildew agent P. xanthii,..
This technology adapts endogenous gene expression…
…P. xanthii and other fungal pathogens is a complicated…..
Therefore, this example set of genes….
The English language in the manuscript is satisfactory overall, but there are a few areas that could benefit from minor editing. There are a few instances in which sentence structure and clarity could be enhanced to improve readability. These alterations would enhance the manuscript's overall coherence and improve the flow of ideas. Therefore, I recommend implementing the suggested minor English revisions to improve the manuscript's clarity and precision.
Author Response
We felt that your comments were very appropriate and helpful to improve the quality of our manuscript. Next, we explained, point-by-point, our responses to your interesting comments, which are also highlights in color through the manuscript:
This review focuses on how fungal infections, particularly Podosphaera xanthii, decrease chitin-triggered immunity in cucurbits, as the abstract indicates. Mentioning the fungal effectors involved in modifying, degrading, or sequestering chitin oligomers demonstrates a thorough comprehension of the subject. The abstract also discusses how this knowledge affects powdery mildew intervention options, which boosts the review's practicality.
Although the ABSTRACT provides a solid overview of the topic and presents the key components of the review, it needs minor improvement. It could be improved by including specific details on the current understanding of chitin-triggered immune signaling and fungal strategies, stating clear objectives or research questions, and limitations and gaps. These changes would improve the abstract and engage readers. It was included.
There are also a few grammatical errors throughout the text, such as:
….two lines of defense: passive and inducible defenses…Done
Remove: for that (……plant immunity, using various strategies such as preventing…..) Removed
…in the modification, degradation, or sequestration…Done
INTRODUCTION
….the rice blast, the gray mold, and the wheat rust, caused by Magnaporthe oryzae, Botrytis cinerea, and Puccinia spp., respectively [2]. Done
…Ug99 race, which is a serious challenge to wheat production. Added
….fungal diseases, powdery mildews take the first position. Done
…a telltale white, dusty coating on leaves, stems, and flowers. Done
They affect several angiosperm plants, including apples, cereals, cucurbits, grapes, peas, tomatoes, and ornamentals such as lilacs, phlox, daisies, and roses [3]. Modified
…is the most important, affecting important…Done
Plants have defense mechanisms to.. Included
..into two categories: The first line..Done
.."passive defenses", which include the cuticle, a waxy covering deposited on the plant surface, the plant cell wall, and anti-microbial substances such as enzymes, peptides, and secondary metabolites [4]. On the other hand, "inducible defense" forms the second line of plant defense mechanisms, which include…Modified
..the infection site, which can eventually lead..included
…as non-self components, …Done
….recognize various microbes via the perception of conserved MAMPs such as lipopolysaccharides from Gram-negative bacteria, peptidoglycans from Gram-positive bacteria, and eubacterial flagellin [9]. Done
hence, chitin does not exist in….Modified
….immunity and pushing the plant to produce further defense…Done
….capable of creating defense….Done
…by plant enzymes or to interfere with….included
…mildew Podosphaera xanthii, will be presented. Done
- CHITIN-TRIGGERED PLANT IMMUNITY
..accumulate into the vacuole and are released upon…Modified
..defense-related gene expression, and even hypersensitivity reactions (Figure 1) Done
Therefore, Chitin Elicitor Receptor Kinase-1 (OsCERK1), a …Done
…recombination, supporting the need for these..Modified
3.1. LYSM AND NON-LYSM CONTAINING EFFECTORS
….in tomato, tobacco, rice cells, and leaves. Included
….callose deposition, and MAPK signaling,…Done
…Moniliophthora perniciosa secretes the inactive chitinase….Included
..the perception of plant chitin receptors and suppress….Done
….and Verticillium dahlia’s Vd2LysM [47], are thought to scavenge chitin fragments and compete with plant chitin receptors. By binding to both chitin polymers and oligomers, they protect the fungal cell wall….Modified
….ChELP1 and ChELP2 are also N-glycosylated [46], indicating that N-glycosylation is…..Done
…Mycosphaerella graminicola), secretes LysM….Done
- CONCLUSIONS
…at two critical moments: during the…Done
A notable example of this issue is the cucurbit powdery mildew agent P. xanthii,..included
This technology adapts endogenous gene expression…Done
…P. xanthii and other fungal pathogens is a complicated…..Done
Therefore, this example set of genes….Done
Comments on the Quality of English Language
The English language in the manuscript is satisfactory overall, but there are a few areas that could benefit from minor editing. There are a few instances in which sentence structure and clarity could be enhanced to improve readability. These alterations would enhance the manuscript's overall coherence and improve the flow of ideas. Therefore, I recommend implementing the suggested minor English revisions to improve the manuscript's clarity and precision. Thank you very much for all these valuables comments. We included all of them. Anyway, the manuscript was edited for proper English language, grammar, punctuation, spelling, and overall style by one or more of the highly qualified native English-speaking editors at the company AJE.

Reviewer 2 Report
The manuscript is a very interesting, complete, and updated review on suppression of chitin-triggered immunity by plant fungal pathogens. The case history reported on Podosphaera xanthii is a well-suited example of what stated in the general part, interestingly performed with the most advanced technologies of molecular biology and bioinformatics. I appreciate very much the clearness of the text and especially of the figures.
Author Response
Thank you very much for your comments. We appreciate it and we are very happy that you liked our manuscript.
